# Guarding Our Vital Systems: A Metric for Critical Infrastructure Cyber Resilience

**DOI:** 10.3390/s25154545

**Published:** 2025-07-22

**Authors:** Muharman Lubis, Muhammad Fakhrul Safitra, Hanif Fakhrurroja, Alif Noorachmad Muttaqin

**Affiliations:** 1Master of Information System Study Program, School of Industrial Engineering, Telkom University, Main Campus (Bandung Campus), Jl. Telekomunikasi No. 1, Bandung 40257, West Java, Indonesia; muhammadfakhrul@telkomuniversity.ac.id (M.F.S.); haniff@telkomuniversity.ac.id (H.F.); weaboo@student.telkomuniversity.ac.id (A.N.M.); 2Department of Network and Security, Pelayaran Nasional Indonesia, Jakarta 10130, Special Capital Region of Jakarta, Indonesia; 3Research Center for Smart Mechatronics, National Research and Innovation Agency, Bandung 40135, West Java, Indonesia

**Keywords:** critical infrastructure resilience, contemporary cyber threats, security evaluation metrics, adoption of advanced technology, IoT, cybersecurity, artificial intelligence

## Abstract

The increased occurrence and severity of cyber-attacks on critical infrastructure have underscored the need to embrace systematic and prospective approaches to resilience. The current research takes as its hypothesis that the InfraGuard Cybersecurity Framework—a capability model that measures the maturity of cyber resilience through three functional pillars, Cyber as a Shield, Cyber as a Space, and Cyber as a Sword—is an implementable and understandable means to proceed with. The model treats the significant aspects of situational awareness, active defense, risk management, and recovery from incidents and is measured using globally standardized maturity models like ISO/IEC 15504, NIST CSF, and COBIT. The contributions include multidimensional measurements of resilience, a scored scale of capability (0–5), and domain-based classification enabling organizations to assess and enhance their cybersecurity situation in a formalized manner. The framework’s applicability is illustrated in three exploratory settings of power grids, healthcare systems, and airports, each constituting various levels of maturity in resilience. This study provides down-to-earth recommendations to policymakers through the translation of the attributes of resilience into concrete assessment indicators, promoting policymaking, investment planning, and global cyber defense collaboration.

## 1. Introduction

In an era of increasing globalization and digitalization dominance, critical infrastructure has undergone significant transformation, evolving into more than just a support system for modern life [1,2]. Today, critical infrastructure is no longer merely a supporting element but rather a complex system that serves as the vital artery, facilitating the life of the economy, maintaining social stability, and safeguarding national security [3,4,5]. These essential networks encompass transportation, energy, clean water, banking, and telecommunications systems, forming the foundation that supports fundamental daily life functions [6]. The security and resilience of critical infrastructure constitute the core driving forces behind the pulse of progress and the resilience of modern society [7,8]. The crucial role of this infrastructure not only influences societal well-being but also economic stability and national sovereignty [9,10]. Therefore, understanding and appreciating the essential role of critical infrastructure in daily life are imperative and cannot be ignored [11,12,13]. However, during an increasingly interconnected and automated modern era, critical infrastructure has become a vulnerable target for sophisticated and concerning cyber-attacks [14,15,16]. Cyber threats are no longer mere speculation; they manifest as real threats capable of disrupting the operations of our crucial infrastructure [17,18,19]. Their impact, akin to shockwaves, can propagate into the economic sector, threaten environmental integrity, and even endanger human lives [20,21,22].

Several notable incidents, such as the Stuxnet attack on Iran’s nuclear facilities [23,24], widespread power outages in Ukraine [25,26], and coordinated cyber-attacks in 2021 targeting water and waste systems in the United States [27,28], all underscore the increased risk to critical infrastructure [29,30]. Therefore, understanding and anticipating these threats are imperative, and the maximum efforts are required to protect critical infrastructure from cyber-attacks. Investment in advanced cybersecurity technologies and strategies is a necessity to safeguard our infrastructure and ultimately ensure the security and well-being of our society. In this context, the Thales 2022 Threat Report for Critical Infrastructure provides a deeper understanding of the impact of cyber-attacks on critical infrastructure [31,32,33]. This report summarizes key findings gathered from surveys of leaders and practitioners in critical infrastructure organizations, offering insights into mitigating risks such as ransomware and malware [34,35]. Interestingly, the survey notes that 79% of the respondents expressed concerns about the security risks of remote work, highlighting how changes in modern work patterns introduce new challenges into the security of critical infrastructure. Equally, 44 percent reported an increase in the volume, severity, and/or scope of cyber-attacks in the last 12 months, with 55% identifying malware as the most common cause of the rise in attacks [36,37]. This emphasizes the escalation of the threats faced by critical infrastructure worldwide.

As a concrete example, Australia reported 143 cyber-attacks on its critical infrastructure in the past year, up from 95 incidents the previous year. These attacks encompassed the energy, utilities, telecommunications, and transportation sectors [38]. Changes in the digital landscape in the last decade are also evident, with previously isolated Operational Technology (OT) systems becoming increasingly connected to the internet [39,40,41]. Smart IoT sensors power water and energy systems, and government operations are deeply rooted in data. Growing dependence on cloud platforms provides a vulnerable attack surface for threat actors and hostile nations [42,43]. In the face of these challenges, understanding and anticipating threats become crucial. The maximum efforts are required to protect critical infrastructure from cyber-attacks. Wise investment in advanced cybersecurity technologies and strategies is a necessity to ensure the security and well-being of our society. Continuous updates and adaptation to developments in technology and cyber threats are crucial steps in ensuring the resilience of critical infrastructure in this increasingly interconnected and automated era.

### 1.1. Research Contributions

Facing the challenges posed by the complexity of cyber threats and the unprecedented pace of technological change, protection and recovery from cyber-attacks have become urgent and inevitable needs. This research aims to design and implement structured strategies and organized methods for evaluating and enhancing cyber resilience in critical infrastructure. An innovative and data-driven resilience model is introduced in this study. Its goal is not only to effectively detect and respond to the impacts on critical infrastructure caused by adversaries but also to estimate and implement proactive preventive measures to prevent system failures and enhance service continuity. Additionally, this research proposes a comprehensive comparative analysis utilizing various cybersecurity metrics and in-depth analyses, leveraging limited features and available data. The objective is to provide better insights into the strengths and weaknesses of current cybersecurity systems and how they can be improved. This research strives to make a significant contribution to the efforts to protect critical infrastructure from increasing cyber threats, ensuring the survival and well-being of society. This includes not only maintaining the integrity and reliability of our infrastructure but also promoting sustainable economic growth and social stability. Therefore, this research contributes to global efforts to create a safer and more prosperous society in the digital era.

### 1.2. Research Questions

This research raises several key questions that are crucial to answer:How can we design and implement an effective constructive resilience model to detect and respond to the impact of cyber-attacks on critical infrastructure?How can we forecast and identify more effective preventive measures to prevent failures in vital infrastructure?How can a comprehensive, in-depth comparative analysis using various metrics but limited features be conducted to measure and enhance the level of security in critical infrastructure?How can this research provide practical and strategic guidance for decision-makers in companies managing critical infrastructure, as well as supporting global efforts to reduce negative impacts and respond to cyber-attacks?

This research aims to address these questions and provide deeper insights into the extent of critical infrastructure’s resilience to cyber threats and how to protect against them. Given the complexity and vulnerability of critical infrastructure to increasingly sophisticated cyber-attacks, this research encourages closer collaboration among various stakeholders, including companies, governments, industry bodies, and non-profit organizations, to strengthen cyber resilience comprehensively [29,44].

### 1.3. Implications and Innovations

In the context of critical infrastructure security, the applied security model plays a crucial role in ensuring comprehensive protection and detecting potential security issues [45,46]. Cyber threats can originate from various sources, including foreign nations attempting espionage operations, financially motivated hacking organizations, and malicious individuals aiming to cause harm [44,47,48]. With the advancement of technology, AI-supported cyber-attacks have become a reality, presenting new challenges and potential solutions to enhance the security of critical infrastructure [49,50]. Therefore, it is important for us to continue adapting and innovating in our security strategies and technologies to keep critical infrastructure safe amidst increasingly sophisticated and diverse cyber threats [51,52]. Overall, this research has very significant implications for efforts to enhance our critical infrastructure’s resilience to increasingly sophisticated cyber-attacks. This research also provides much-needed practical guidance on protecting our society and economy from cyber threats that can shake the foundations of our existence. This is a crucial step in maintaining quality of life, social stability, and economic sustainability in the face of cyber threats in this digital era. This research serves as a crucial pillar in building our defense against cyber threats and ensuring that our society and economy remain safe and thriving in this digital era. By leveraging the latest security technology and innovative defense strategies, this research aims to build a robust and adaptive security system that can respond to and counter increasingly complex and sophisticated cyber threats. Additionally, this research seeks to understand and anticipate the evolution of future cyber threats, ensuring that we can continue to protect our critical infrastructure and ensure the survival and well-being of our society.

## 2. Related Studies

### 2.1. A Journey Through Prior Works

In confronting escalating cyber threats to critical infrastructure, several studies have made significant contributions to understanding and addressing these challenges. These studies encompass various aspects, ranging from the utilization of artificial intelligence (AI) to detect and counter threats to the development of community sustainability models and practical tools that assist organizations in more effectively facing cyber threats. Therefore, it is crucial to be aware that diverse and innovative approaches are highly necessary to tackle increasingly complex and dynamic cybersecurity challenges.

One of the primary contributors to our understanding and handling of cyber threats to critical infrastructure is Abuhasel’s study [9]. In his work, Abuhasel proposed the Constructive Resilience Model Induced by Artificial Intelligence (AI-CRM) as a progressive step to enhance the cybersecurity of critical infrastructure. This model not only considers the potential influences that adversaries may have on infrastructure elements but also calculates probabilities based on the impact of previous attacks on infrastructure failures and responses to operational service. Through this approach, resilience can be improved by adding security measures that respond to the impact of attacks. Prokhorenko and Babar [53] also contribute by proposing a comprehensive architectural approach to enhancing the resilience of cloud-, Fog-, and Edge-based systems in the context of critical infrastructure. They introduce a capability-based framework designed to strengthen overall system resilience. Besides addressing trust issues in the context of resilience and system reliability, this research provides in-depth insights into existing solutions to enhance the resilience of distributed systems. Carías et al. [54] take a practical approach by developing a web-based operational tool to help organizations operationalize cyber resilience in critical infrastructure. This tool not only provides organizations with the ability to follow a comprehensive process, including the implementation of a cyber resilience framework, but also integrates a Cyber Resilience Self-Assessment Tool (CR-SAT) tested through case studies. Thus, this research highlights how web-based tools and technologies can facilitate and strengthen cyber resilience within organizations. Clark and Zonouz [55] discuss the robust operation of cyber-physical infrastructure in potentially adversarial environmental situations. They present a formal definition of resilience and assessment metrics that measure the system’s ability to recover from attacks within a specific time interval and the recovery cost. This approach illustrates how resilience assumes that sophisticated attacks can bypass protection and detection mechanisms, and thus, a robust system must be able to respond through reactive and proactive attack tolerance mechanisms. Valinejad and Mili [14] bring the concept of community sustainability into the understanding of resilience. They design a multi-agent model that integrates cyber, physical, and social aspects to understand readiness and adaptability in the face of threats. This research highlights that cooperation within a community can have a significant positive impact on individual behavior and that strong relationships within the community are a key factor in strengthening resilience. Domínguez-Dorado et al. [56] emphasize the importance of implementing a reference model in managing cybersecurity at a lower level in critical infrastructure. They propose a process which they call CyberTOMP for managing cybersecurity at this level and provide methodological elements supporting its implementation. This research indicates that cybersecurity management requires an integrated and holistic approach. Kumar, Alvarez, and Kumar [57] conducted relevant research on security resilience in commercial smart devices in the context of critical infrastructure. Their study discusses how cybersecurity attacks can affect the operations and data integrity of these smart devices, which play a central role in an increasingly important smart grid network. Ashley et al. [6] sound the alarm about the urgency of cybersecurity in critical infrastructure and introduce the Network Defense Training Game (NDTG) as a cybersecurity training platform. The NDTG uses scenario-based narratives based on historical cyber incidents and is designed to train users in their understanding of and skills in handling cybersecurity events and incidents in critical infrastructure. Makrakis et al. [45] provide a comprehensive survey of the threats and attacks on industrial control systems and critical infrastructure. This survey provides an in-depth understanding of the various threats and vulnerabilities faced by critical infrastructure. Simone et al. [58] present an innovative approach that combines the STAMP model with System-Theoretic Process Analysis for Security (STPA-Sec) and simulations to identify vulnerable controls across socio-technical systems in the context of critical infrastructure. This method has been applied in a case study on a water treatment plant and helped improve the resilience of that infrastructure to cyber threats.

Overall, these studies contribute significantly to understanding and enhancing resilience and cybersecurity in the context of critical infrastructure. With diverse approaches, ranging from the AI-CRM model to game-based cybersecurity training, as well as an analysis of cyber-attacks on smart meters and industrial control systems, these studies aim to strengthen the resilience of critical infrastructure against evolving threats. These studies play a key role in maintaining the continuity of operations and the security of systems vital to modern society and the economy. This demonstrates that research and innovation in this field are crucial to ensuring the security and resilience of our critical infrastructure.

### 2.2. Developing a Metric for Critical Infrastructure’s Cyber Resilience

Given the importance of maintaining the security and resilience of critical infrastructure in the face of increasingly complex cyber threats, the next step is to delve deeper into the development of a metric that can be used to measure the level of cyber resilience in critical infrastructure [59,60,61]. This metric plays a central role in assisting stakeholders, from companies to governments and non-profit organizations, in measuring the extent to which they have protected their infrastructure and how they can recover in emergency situations caused by increasingly complex cyber-attacks [62,63]. Additionally, this metric is crucial in evaluating the effectiveness of the strategies and tactics used to protect critical infrastructure, as well as in determining necessary improvement measures [64].

#### 2.2.1. The Essence of a Critical Infrastructure Cyber Resilience Metric

A metric for measuring the cyber resilience of critical infrastructure should accurately reflect its complexity and vulnerability to evolving cyber threats [65,66]. Given the vital role of critical infrastructure in maintaining social and economic stability, this metric should provide a deep understanding of how stakeholders, including companies, governments, and non-profit organizations, can protect their infrastructure and to what extent they can recover in emergency situations [67]. This metric becomes a crucial cornerstone in ensuring that critical infrastructure continues to operate efficiently and securely, as well as in evaluating the effectiveness of the strategies and tactics used to protect critical infrastructure [68,69,70]. Furthermore, this metric serves as an objective and reliable evaluation tool for determining necessary improvement measures [71,72].

#### 2.2.2. The Key Components of a Critical Infrastructure Cyber Resilience Metric

There are several key components of assessing critical infrastructure:Risk Assessment and Vulnerability Assessment [73]: This metric should include a comprehensive risk assessment to identify potential vulnerabilities in critical infrastructure. This involves evaluating potential cyber threats that infrastructure may face, how often they may occur, and their potential impact [74]. In this context, providing vulnerability scores and risk levels may be necessary to help organizations prioritize actions and allocate resources [75]. Additionally, this assessment should also consider factors such as existing security policies, implemented security controls, and the organization’s level of readiness to face cyber threats [76].Response and Recovery Capabilities [77]: This metric should reflect the extent to which critical infrastructure can respond to and recover from cyber-attacks. This includes assessing readiness for cyber incidents, including personnel training, emergency plans, and tools and systems supporting recovery [78]. The ability to respond quickly and effectively to attacks and recover efficiently is a key element in enhancing cyber resilience. Additionally, this metric should also consider factors such as response times, the effectiveness of response actions, and recovery process efficiency.The Use of Advanced Technology [79]: The use of advanced technology, such as artificial intelligence, data analytics, and early detection systems, should be measured using this metric [80]. The implementation of these technologies can significantly enhance detection and response capabilities in the face of increasingly complex cyber threats. Therefore, evaluating the use of innovative technology is a key component of a metric of cyber resilience. Additionally, this metric should consider the extent to which these technologies have been integrated into the organization’s security system and their effectiveness in detecting and responding to cyber threats [81].Collaboration and Information Sharing [82]: The level of collaboration with other stakeholders, both in the public and private sectors, should be a significant assessment factor in this metric [83]. The ability to share information and collaborate in protecting critical infrastructure collectively is a crucial element in ensuring optimal cyber resilience [84,85]. Additionally, this metric should also consider the extent to which organizations have built and maintained effective collaborative relationships with other stakeholders.Impact Measurement and Recovery Times [86]: This metric should include measurement of the impact of cyber-attacks on critical infrastructure, including how quickly infrastructure can recover and resume operations after an attack [87,88]. This measurement provides insights into how much attacks affect operations and how long it takes for full recovery. Additionally, this metric should also consider factors such as the financial impact of attacks, the impact on the organization’s reputation, and the impact on customers or users of critical infrastructure.

#### 2.2.3. The Metric Development Process

The process of developing this metric is a systematic and ongoing step that involves several crucial stages:Analysis of Cyber Threat Trends and Patterns [89]: Focus on analyzing trends and patterns in cyber threats without having to collect specific data [90]. Identify common characteristics of previous attacks, frequently targeted infrastructure, and relevant performance indicators. By understanding these trends, strategic insights can be developed to enhance security without further data collection [89].The Implementation of Security Metrics [91]: Focus on implementing security metrics by utilizing all previously identified information. This involves applying formulas or equations designed to measure security based on identified trends and patterns in cyber threats [92]. Additionally, the metric implementation should consider factors such as the metric’s consistency with organizational goals, the metric’s usability, and the metric’s capability to provide significant insights.Contextual Analysis and Metric Assessment [66,93]: At this stage, the focus is on a contextual analysis and metric assessment. This involves exploring real-world situations without relying on testing [45]. The process may include narrative-based assessments related to the effectiveness of the metric in depicting potential cyber threats faced by organizations. Metric evaluation also needs to consider the reliability, consistency, and relevance of the metric to security environmental dynamics.Advanced Development [94]: This stage involves reviewing and improving the metric based on testing results and feedback from users. This may involve adjusting formulas or equations, adding or subtracting metric components, or changes in the data collection processes. The goal is to ensure that the metric remains relevant and effective in measuring the cyber resilience of critical infrastructure [71].

#### 2.2.4. Applications of the Metric

The metric of cyber resilience in critical infrastructure has several significant applications:An Evaluation of Current Resilience Levels [95]: Organizations can use this metric to evaluate the current level of cyber resilience and identify areas where improvements are needed [96,97]. This helps organizations identify priorities for improvement actions and plan effective strategies for enhancing cyber resilience.Comparison with Standards and Regulations [98]: This metric allows organizations to compare their level of resilience with industry standards or applicable regulations. This helps organizations ensure compliance with existing guidelines and regulations and determine areas where they may need to make improvements.Planning and Resource Allocation [99]: This metric helps organizations plan efficient resource allocation efforts to improve cyber resilience [100]. With the data generated by the metric, organizations can prioritize their resource usage and plan effective strategies for improving cyber resilience.Reporting and Accountability [101]: This metric can be used to provide reports on the status of cyber resilience to stakeholders and regulatory authorities [102]. This is a crucial step in maintaining transparency and accountability regarding the security of critical infrastructure.Comparison with Other Organizations [103]: This metric allows for comparison with other organizations in the same sector or a similar sector. Through this comparison, organizations can identify the best practices and see how they compare with other organizations.

The development of a metric for cyber resilience in critical infrastructure is a crucial step in safeguarding increasingly interconnected critical infrastructure from complex cyber threats. This metric provides a clear insight into how well critical infrastructure is protected and can recover in emergency situations. Thus, this metric is not only a measurement tool but also a vital instrument in ensuring the continuity of operations of critical infrastructure crucial to the well-being and sustainability of our society and modern economy. In the face of evolving cyber threats, the development of this metric is a proactive step in ensuring the resilience of critical infrastructure crucial to our well-being and sustainability.

### 2.3. Fortifying Resilience in Critical Infrastructure

To comprehend the intricacies of critical infrastructure resilience and why it is crucial in an increasingly interconnected and complex world, we need to delve deeper into this concept [104,105]. In the context of critical infrastructure, resilience refers to a system’s ability to withstand and function in accordance with its primary objectives [106,107]. For instance, when considering backup solutions, the primary goal is to maintain the system’s reliability in the face of potential data losses, which could be a significant disaster if not handled properly. This is why resilience becomes key in safeguarding the operations of critical infrastructure, often under high pressure [54,108].

#### 2.3.1. Resilience vs. Reliability: Differences and Interconnection

It is essential to differentiate between “resilience” and “reliability”, as these terms are often used interchangeably [109]. In the context of critical infrastructure, resilience is, in fact, a key element in achieving a high level of reliability. Therefore, while resilience is the process that enables infrastructure to continue operating even in challenging conditions, reliability is the desired outcome of that process [110]. We can view resilience as the foundation that enables reliability. In many ways, resilience is crucial to achieving the desired reliability of critical infrastructure systems.

While conceptually different, resilience and reliability also differ quantitatively. In this context, resilience may be defined as a function of the primary reliability measures. For instance, resilience (R) may be approximated as(1)R=(1−MTTR/MTBF)
where MTTR stands for the Mean Time to Recovery and MTBF stands for the Mean Time Between Failures. The equation suggests that decreased recovery times and increased operational stability contribute to greater overall resilience. Although no empirical evidence is included in this present study to validate this formula, the equation offers the basis for subsequent simulations and field testing. Follow-up research could analyze historical incident records with the aim of calibrating and implementing this measure for different kinds of infrastructure.

#### 2.3.2. Addressing Diverse Challenges

Critical infrastructure faces various challenges that can impact its operations. Physical threats such as natural disasters, terrorist attacks, and increasingly sophisticated cyber threats are concrete examples of these challenges. In operational reality, resilience solutions often focus on groups of potential factors that may disrupt infrastructure. By understanding these factors deeply, we can develop more effective and reliable resilience strategies to address these diverse challenges.

#### 2.3.3. Fault Tolerance Mechanisms: The Heart of Critical Infrastructure’s Resilience

In the context of critical infrastructure, fault tolerance mechanisms are a key element that enables reliable and continuous operations. These mechanisms consist of two main layers. First, there is the business logic layer responsible for maintaining the normal operation of the system, ensuring that all operations proceed as planned. Then, there is the meta-layer that handles errors and the recovery process in case of disruptions [53,111]. This concept allows for flexibility in implementing fault tolerance in existing or developing infrastructure [112]. Moreover, it is important not only to focus on the resilience of physical infrastructure but also on the resilience of agents or monitoring modules, often overlooked in resilience planning. Some approaches even emphasize what is referred to as “gray failure” rather than “hard failure”. This means that the primary focus is on resilience to specific service disruptions rather than just focusing on potential failures [113,114,115].

#### 2.3.4. Categories of Resilience Solutions for Critical Infrastructure

If we categorize resilience solutions for critical infrastructure, we can classify them into three main categories, as depicted in Figure 1:

Cyber as a Shield [116]: This category includes various crucial aspects such as situational awareness, security assurance, and resilience principles. The focus of this category is to protect systems and data from various threats and attacks [117,118]. Situational awareness involves understanding and knowledge of the operational system environment and how changes in the environment can affect system operations. Security assurance involves a series of steps taken to ensure that the system is protected from various threats and attacks. Resilience principles refer to fundamental principles guiding the design and operation of systems to ensure their resilience to threats.

Cyber as a Space [116]: The second category includes aspects such as risk management, infrastructure resilience, and infrastructure readiness. The focus of this category is to ensure that critical infrastructure can operate effectively and efficiently in various conditions. Risk management involves identifying, assessing, and prioritizing risks, followed by resource allocation to minimize, monitor, and control risk impacts. Infrastructure resilience refers to the ability of infrastructure to withstand and recover from various threats and challenges. Infrastructure readiness involves pre-emptive steps to prepare infrastructure for potential threats and attacks.

Cyber as a Sword [116]: The third category includes aspects such as active defense, critical infrastructure awareness, infrastructure protection policies, and critical incident recovery. The focus of this category is to take proactive actions in detecting, preventing, and responding to attacks on systems. Active defense involves proactive actions taken to detect, prevent, and respond to attacks on systems. Critical infrastructure awareness involves an understanding of the importance of critical infrastructure and how vulnerabilities in that infrastructure can affect national security and the economy. Infrastructure protection policies involve policies and procedures designed to protect infrastructure from physical and cyber threats. Critical incident recovery involves the steps taken after an incident to restore normal system operations as quickly as possible.

A model known as the “InfraGuard Cybersecurity Framework” is formed by these three main categories of critical infrastructure resilience solutions (Table 1). This model provides a clear and comprehensive view of the framework’s overall structure. By reviewing this model, we can understand how each part of the framework interacts and works together to form a robust and efficient security system. Figure 2 displays this model. It is important to remember that critical infrastructure is the backbone of our daily lives, both economically and in terms of national security. With a profound understanding of resilience concepts in the context of critical infrastructure, we can prepare better to face various potential threats. With the increasing interconnectedness of the world and the growing complexity of cyber-attacks, effective resilience strategies have become increasingly vital in maintaining operational continuity and societal security. By continuously developing and implementing appropriate resilience solutions, we can enhance the resilience of our critical infrastructure and protect our future [119]. Furthermore, understanding the concept of resilience also helps us appreciate the role of critical infrastructure in maintaining the stability of our social and economic life, ultimately influencing our quality of life [120,121]. Therefore, preparing critical infrastructure well is a shared responsibility for all of us. As we move towards an increasingly complex future, resilience in critical infrastructure is a strong foundation that will help us face challenges with confidence and self-assurance [122,123].

## 3. Method

### 3.1. Charting Resilience Metrics

#### 3.1.1. The Important of Process Evaluation Models in Resilience Measurement

To gain a deeper understanding of the concept of resilience in the context of critical infrastructure, it is necessary to explore process evaluation models [124,125]. Process evaluation models form a vital framework in the detailed understanding and measurement of resilience, and this becomes a key element in efforts to enhance the resilience of critical infrastructure [126,127]. This model consists of two main dimensions. The first dimension is the type of process. This dimension refers to various types of processes related to critical infrastructure. These processes are detailed and grouped into meaningful categories. For example, this includes cybersecurity processes, disaster recovery processes, or monitoring and response processes.

The “InfraGuard Cybersecurity Framework” visual model is newly synthesized by the authors to quantify and define cyber resilience for critical infrastructure contexts. The model offers a two-dimensional methodology through the crossing of capability maturity levels (Level 0 to Level 5) and functional domains categorized as Cyber as a Shield, Cyber as a Space, and Cyber as a Sword. These metaphorical spaces are referred to individually to encapsulate the individual strata of organizational protection across passive safeguarding and system robustness to active incident management. While the constituent components (for example, situational sensing, infrastructure robustness, incident restoration, etc.) are widely recognized in the cybersecurity literature, the nomenclature, configuration, and diagonal matrix representation are new to this research.

This framework takes conceptual cues from several recognized models and standards. These vertical maturity levels are based on concepts in capability maturity models such as ISO/IEC 15504 (now ISO/IEC 33000 series) and the COBIT Process Assessment Model, which outline process sophistication and institutionalization levels. The horizontal classification of elements reflects themes in the NIST Cybersecurity Framework (Identify, Protect, Detect, Respond, Recover), rearranged under a new taxonomy. In addition, the model encourages the philosophy of use cases such as AI-CRM, STPA-Sec, and DHS’s Cyber Resilience Review, which emphasize the utilization of stacked, dynamic defense systems. By combining these backgrounds into one consistent and easy-to-understand model, this framework is an easy tool to employ to assess, communicate, and guide cyber resilience strategy enhancements.

The second dimension in this model is the level of competence. In this dimension, process attributes are grouped into several competence levels. These attributes are measurable characteristics that help us classify how effective and competent a process is in performing its tasks. These attributes may include the process’s ability to detect threats, respond to incidents, or plan for recovery.

#### 3.1.2. Levels of Process Competence in the InfraGuard Cyber Resilience Framework

The “InfraGuard Cyber Resilience Framework” is a crucial tool that helps categorize security processes into six levels based on their progress and effectiveness. How resilience metrics are charted can be observed in the illustration in Figure 3.

Level 5—Security Process Enhancements: This is the highest level in the framework, where security processes have been fully optimized. At this stage, security processes not only run smoothly but are continually improved and perfected based on feedback and learning from previous experiences. Organizations at this level have achieved the highest maturity level in security processes, and each aspect is optimized for full effectiveness. Organizations at this level are leaders in cybersecurity practices.Level 4—A Consistent Security Process: At this level, security processes run with consistency and predictability. The processes yield consistent results and meet established quality standards. Consistency is key here, meaning organizations can rely on security processes to deliver predictable results without much variation or uncertainty. Organizations at this level have achieved a very high level of resilience in maintaining cybersecurity.Level 3—A Solidified Security Process: At this level, security processes have become robust and established. The processes have proven effective in practice and have become an integral part of daily operations. This indicates that organizations have successfully built a strong foundation for their security, and these processes are considered mature practices in their operations. Organizations at this level have achieved a high level of resilience in maintaining their critical infrastructure.Level 2—A Supervised Security Process: At this level, security processes are closely monitored to ensure that all activities proceed according to the planned and established standards. Supervision is a crucial component here, and organizations ensure that security processes unfold as expected, although there may still be room for improvement. Organizations at this level are striving to enhance their resilience and planning the necessary steps to reach a higher level.Level 1—An Executed Security Process: At this level, security processes are being executed. Basic security measures have been implemented, and the processes are running according to the basic plan. This is an initial step indicating that the organization has taken basic actions to protect its infrastructure. While work still remains, the first step towards resilience has been taken.Level 0—An Unfinished Security Process: This is the lowest level in the framework, where the security processes are unfinished. Some aspects of the processes may not have been implemented or may not be functioning well. This indicates that significant work is needed to achieve a worthy level of resilience.

#### 3.1.3. The Social Dimension in Resilience Measurements

Not only processes and technology influence the resilience of critical infrastructure. Social factors also play a crucial role in how people interact and collaborate in the face of threats [128]. The level of understanding and awareness in society about cybersecurity threats significantly impacts the ability to contribute to maintaining the security of critical infrastructure. The higher this level of understanding and awareness, the better society can participate in ensuring security [129]. This understanding includes knowledge of potential threats and the actions to be taken in emergency situations. Coordination and collaboration among various stakeholders also become crucial elements in resilience [130]. The ability to work together and share information effectively can enhance responses to threats and help ensure the continuity of critical infrastructure operations.

#### 3.1.4. Integrating Process and Social Dimensions into Resilience Measurements

By combining process and social dimensions into resilience measurements, we can understand how complex critical infrastructure faces challenges [131,132]. Comprehensive measurement involves evaluating organizational processes and public understanding, as well as the readiness and coordination levels among various stakeholders. In this way, we can obtain a deeper insight into the level of resilience of critical infrastructure and identify areas where improvements are needed.

#### 3.1.5. The Importance of a Multidimensional Approach

The conclusion that can be drawn is that measuring the resilience of critical infrastructure is a crucial step in ensuring cybersecurity. In an era of increasingly complex threats, there is no single solution that can guarantee resilience. Instead, a multidimensional approach that includes a deep understanding of organizational processes, the level of competence in performing security tasks, and the role and readiness of society is key to achieving higher resilience levels [133]. All parties involved in critical infrastructure, whether government, private organizations, or the public, need to contribute to these efforts to ensure uninterrupted operational continuity. By focusing on comprehensive measurement and understanding, we can build a safer future that is more resilient to evolving cybersecurity threats. Through joint efforts and collaboration, we can maintain the resilience of critical infrastructure, which, in turn, will help protect the social and economic stability we heavily depend on. In doing so, we can move forward in facing future challenges that are unseen but are sure to come.

### 3.2. The Resilience Spectrum

#### 3.2.1. The Conceptual Structure of the Resilience Spectrum

In an era characterized by constant change and uncertainty in the cyber world, understanding the spectrum of resilience is a crucial obligation to secure critical infrastructure and processes. This involves evaluating the ability of a process to achieve business goals, both ongoing and expected in the future. The capability dimension is a key parameter in measuring the resilience level of processes or infrastructure. This model considers a set of process attributes grouped into various capability levels, following the guidelines provided by ISO/IEC 15504-2:2003 and COBIT 5. We will classify the capability indicators for each aspect: Cyber as a Shield, Cyber as a Space, and Cyber as a Sword [116].

Additionally, the resilience spectrum is represented in Figure 4, providing a visual depiction of the spectrum’s significance in enhancing the resilience of critical systems.

Cyber as a Shield [116]:Situational Awareness: Situational awareness is crucial in enhancing resilience to cyber threats. Organizations and infrastructure need to actively monitor and detect changes in their operational environment that can affect security. This includes monitoring network traffic, analyzing suspicious activities, and understanding current trends in cyber-attacks.Security Assurance: Security assurance involves the actions taken to ensure that systems are secure from threats and attacks. This includes periodic security risk assessments, the implementation of risk mitigation measures, and ensuring strict adherence to security standards.Active Defense: The ability to conduct active defense is crucial. Organizations need to have strategies and tools that allow them to detect attacks as early as possible, respond quickly, and even take proactive actions to block or thwart attacks before they damage the system.Risk Management: Effective risk management is a vital component of cyber defense. This involves identifying, assessing, and managing the risks associated with cyber-attacks. Risk management can help organizations prioritize risk mitigation and allocate appropriate resources.

Cyber as a Space [116]:Infrastructure Resilience: The ability of infrastructure to withstand and recover from various threats is key to creating a safe space in the cyber context. This involves planning and implementing strategies to maintain infrastructure operability even in challenging situations.Critical Infrastructure Awareness: Awareness of the importance of critical infrastructure is the first step in protecting it and mitigating risks. Organizations need to understand the vulnerabilities in critical infrastructure and how its vulnerabilities can impact national security and the economy.Resilience Principles: Resilience principles should guide the design and operation of infrastructure. This includes fundamental principles that guide the design, implementation, and maintenance of infrastructure to remain resilient to threats.Infrastructure Safeguarding Policy: Policies and procedures designed to protect infrastructure from physical and cyber threats are key steps in creating a safe space in the cyber context.

Cyber as a Sword [116]:Infrastructure Preparedness: Infrastructure preparedness involves pre-emptive measures to prepare infrastructure for potential threats and attacks. This includes planning, personnel training, and security scenario testing.Critical Incident Recovery: Taking steps after a cyber incident to restore normal system operations as quickly as possible is crucial. This involves system recovery, data recovery, and measures to avoid similar incidents in the future.

In the ever-changing cyber world, a deep understanding of the resilience spectrum is key to protecting infrastructure and ensuring business continuity. Organizations must continually adapt and innovate to face evolving challenges. With a deeper understanding of each aspect in the resilience spectrum, organizations can identify areas that need improvement and further investment, ensuring they can maintain their resilience in a dynamic cyber world. As cyber-attacks increase and risks continue to evolve, understanding the resilience spectrum is a valuable guide to keeping critical infrastructure and processes secure and high-performing.

#### 3.2.2. A Quantitative Scoring Model for Resilience Components

To enable the proposed framework to be applied in practice, a quantitative scoring system is introduced for each element of the InfraGuard Cybersecurity Framework. The system enables organizations to measure their cyber resilience maturity in different functional areas in a structured and measurable way. By translating qualitative assessments into numerical scores, stakeholders can more readily understand existing gaps, monitor improvements over time, and support decision-making with evidence-based metrics. Each element in the model, from situational awareness to incident recovery, is assigned a set of definite indicators that reflect its operational maturity. It is rated on a 0 to 5 scale, which indicates increasing levels of maturity, from non-existent or ad hoc processes to fully integrated, automated, and optimized capabilities. The rating is based on definite criteria, such as monitoring coverage, system availability, policy enforcement, and recovery time, designed to be realistic and quantifiable. The framework allows technical teams, not to mention management, to align their measurement activities with global standards such as ISO/IEC 15504, NIST CSF, and COBIT. The scoring values assigned to each component in Table 2 are designed to reflect measurable indicators based on the available operational data. These values may be obtained through methods such as system log audits, uptime/downtime tracking, training records, and structured expert assessments, depending on the organization’s internal monitoring capacity. The resulting component-level scores contribute to the overall resilience evaluation, as illustrated in Table 3, which aggregates these scores into a total resilience level classification.

This quantitative model is not only a benchmarking tool but also a roadmap to incremental cybersecurity improvements for critical infrastructures. By identifying weak areas and defining measurable goals, organizations can prioritize actions based on their urgency and available resources. Furthermore, this scoring system can support internal auditing, policy reviews, and investment decision-making, as well as cross-organization comparisons and reporting to regulatory bodies. Last, the table offers a brief, adaptable template for translating complex notions of resilience into specific actions.

## 4. Results

### Performance Metrics

Performance metrics, as shown in Figure 5, are used to evaluate whether process attributes have been achieved.

To measure an organization’s or infrastructure’s ability to face cyber threats according to three main aspects, Cyber as a Shield, Cyber as a Space, and Cyber as a Sword [116], the following are assessment indicators that can be used to provide a deeper understanding of how an organization or infrastructure assesses its readiness to face cyber threats:

**Level 5—Security Process Enhancements**: At Level 5, organizations have reached their peak readiness in facing increasingly complex cyber threats. “Enhanced Security Process”, as a central element at Level 5, reflects a high level of adoption of the latest technology. Organizations at this level not only keep up with the latest developments but also actively innovate. In addition to adopting the latest technology, they also implement continuous improvement measures, including comprehensive updates to technology, policies, and security practices. Security processes at Level 5 are a perfect blend of cutting-edge technology and continuous optimization. “Upgraded Security Process” emphasizes that security processes have been substantially enhanced in terms of their effectiveness, efficiency, and reliability. Organizations at Level 5 have successfully created security processes that operate with a high level of reliability, consistently and efficiently responding to threats. They have implemented technology updates and improved their security policies comprehensively. Furthermore, “Enriched Security Process” sharpens the focus on a deep understanding of cyber threats. Organizations at Level 5 not only rely on high-end technology but also deepen their understanding of various threats. They engage various stakeholders and apply a holistic approach to combating cyber-attacks. Their security processes not only mitigate risks but also provide valuable insights for executive decision- making. Organizations at Level 5 serve as examples of innovation, adaptation, and leadership in facing cyber threats.

**Level 4—Consistent Security Processes**: Level 4 emphasizes consistency and reliability in executing security processes. “Consistent Security Process” indicates that organizations can routinely carry out security actions and produce stable results. They can consistently respond to threats and predict their outcomes. Organizations at Level 4 have achieved remarkable discipline and alignment in executing security processes. They execute security actions routinely, creating a very high level of readiness. “Stable Security Process” indicates that security processes operate with stability and consistency. They have successfully maintained the reliability of their security processes in the face of changes in the operational environment. Organizations at Level 4 have reached a stage where their security processes remain effective even in the face of rapidly evolving cyber threats. “Regular Security Process” reflects an organization’s discipline in executing security processes according to the existing policies and guidelines. They have ensured that each step is closely followed according to established procedures. Additionally, Level 4 is a stage where organizations have successfully struck a balance between flexibility and discipline in executing security processes. They can adapt to new threats and respond with high consistency. Organizations at Level 4 are exemplars of discipline, consistency, and reliability in facing cyber threats.

**Level 3—Solidified Security Processes**: Level 3 is a stage where organizations have successfully solidified their security processes as an integral part of their day-to-day operations. “Solid Security Process” indicates that security processes have become solid and established. These processes have proven effective in protecting data, systems, and organizational operations. They have successfully created a strong and proactive security culture throughout the organization. Organizations at Level 3 have a very high level of readiness to face cyber threats. “Firm Security Process” indicates that security processes function securely and provide reliable protection against cyber-attacks. Organizations at Level 3 can confront attacks with confidence that their security processes will preserve the integrity, confidentiality, and availability of their information. They have built a strong foundation for maintaining cybersecurity throughout the organization. “Steady Security Process” is a stage where security processes operate stably and can handle threats effectively. They have achieved a balance between responding to cyber threats and day-to-day operations. Organizations at Level 3 are role models in integrating security into every aspect of their operations, resulting in strong resilience against cyber threats.

**Level 2—Supervised Security Processes**: At Level 2, tight supervision takes the spotlight. “Supervised Security Process” indicates that organizations ensure that all activities proceed according to established plans and standards. With strict supervision, organizations can ensure discipline in executing security processes. This supervision includes monitoring activities and ensuring that actions comply with established plans and standards. Organizations at Level 2 have a robust supervision system that ensures consistency in executing security processes. “Monitored Security Process” notes that security processes are regularly monitored to detect anomalies or policy violations that may occur. This active supervision allows organizations to detect issues quickly and respond accordingly. Organizations at Level 2 have achieved a high level of supervision, enabling them to identify and address potential risks effectively. “Controlled Security Process” emphasizes taking security measures with tight controls according to existing guidelines. In this context, control is crucial to ensuring that security processes run as expected. Organizations at Level 2 have achieved a high level of supervision and tight control in executing their security processes.

**Level 1—Executed Security Processes**: At Level 1, organizations have implemented basic security measures and executed them according to basic policies. This process represents the initial steps in building the foundation for higher levels of security. They have embarked on their journey toward higher security levels. “Executed Security Process” indicates that organizations at Level 1 have implemented basic security measures effectively. They execute security processes according to basic policies and established guidelines. Although they are still in the early stages of the journey towards higher readiness to face cyber threats, these initial steps demonstrate their commitment to protecting their assets and data. “Operated Security Process” reflects that organizations at Level 1 execute security processes according to basic policies, although they have not yet achieved a high level of consistency. This is the initial stage of building a strong foundation in executing security processes. They have started their journey to enhance their security, but more time and effort are needed to reach higher levels. “Run Security Process” shows that security processes are being executed, albeit still in the early stages of development and implementation. Organizations at Level 1 have taken the first steps in their journey toward higher security levels. They have initiated efforts to improve their security but still require further development and a deeper understanding of cyber threats.

**Level 0—Unfinished Security Process**: At Level 0, organizations are aware that their security processes are unfinished and require further planning and actions for implementation. This process is still in the early stages of design and needs a deeper understanding of the cyber threats faced. Organizations at Level 0 are aware that they need to start their journey to facing cyber threats and plan the steps they will take. “Unfinished Security Process” reflects that organizations at Level 0 have identified that further efforts are needed to enhance their security. Their security processes are still in the early stages of design and have not been fully implemented. This is a call for improvements and developments in executing security processes. “Incomplete Security Process” indicates that some aspects of security processes may not have been implemented or may not be functioning well. Organizations need more effort to address these weaknesses and ensure that their security processes are more complete. “Partially Done Security Process” suggests that some security steps may have been taken, but these processes are still far from the expected effectiveness. Organizations at Level 0 have initiated efforts to improve their security but still require further development and a deeper understanding of cyber threats.

The level of readiness to face cyber threats is an integral component in maintaining the security and continuity of organizational operations, infrastructure, and information systems in the current digital era. This readiness encompasses several crucial aspects that form the foundation of defense and resilience against increasingly complex cyber-attacks. One major aspect of readiness for cyber threats is technology adoption. Organizations at the forefront of readiness can adopt the latest technology in the context of cybersecurity. They implement cutting-edge security solutions and tools that help them identify, mitigate, and respond to threats effectively. The adoption of the latest technology also includes continuous updates and monitoring of new developments in security technology. Organizations that can follow security technology trends have an advantage in facing constantly evolving cyber threats. In addition to technology adoption, consistency in executing security processes is a key factor in readiness. Consistency involves the routine and predictable implementation of security actions. Organizations with consistent security processes execute them with a high level of order, producing stable results. Consistency also includes maintaining the reliability of security processes in the face of changes in the operational environment. High consistency in executing security processes creates a high level of trust in protecting assets and data.

Supervision is another crucial aspect of readiness for cyber threats. Organizations that implement strict supervision ensure that all activities proceed according to established plans and standards. With tight supervision, organizations can ensure that security processes are executed with discipline and in accordance with existing guidelines. This supervision includes monitoring security activities to detect anomalies or policy violations that may occur. Organizations that can detect issues quickly and respond accordingly have an advantage in facing cyber threats. The integrity of security processes is a key foundation in readiness for cyber threats. Organizations that ensure the integrity of security processes execute procedures and policies strictly, preventing violations or manipulations that attackers may attempt. Integrity also includes a deep understanding of cyber threats and the maintenance of strong security principles. Organizations with uncompromising security processes have a stronger defense against cyber-attacks [134,135].

Adaptation and innovation are also essential elements in readiness for cyber threats. Organizations that can adapt quickly to new threats and innovate their security solutions have an advantage in facing increasingly complex attacks [136,137]. The ability to respond to cyber threats with flexibility and creativity allows organizations to stay ahead in the battle against cyber-attackers. In addition to technical aspects, a security culture also plays a crucial role in readiness. Organizations that create a strong security culture encourage all team members to prioritize security in every action and decision they make. A security culture shapes a proactive attitude toward facing cyber threats and turns security into a shared responsibility. With a strong security culture, organizations can create more effective defenses against cyber-attacks. Periodic readiness evaluations are an important tool in helping organizations identify their level of readiness in facing cyber threats. By understanding how far they have progressed in each aspect of readiness, organizations can plan and implement continuous improvements. Periodic evaluations also allow organizations to monitor their progress in enhancing their level of cybersecurity readiness. A combination of technology adoption, consistency, supervision, integrity of security processes, adaptation, innovation, and a security culture forms a comprehensive framework in facing cyber threats [138,139]. Organizations that combine these elements well have a strong defense against constantly evolving cyber threats. Strong cybersecurity readiness not only secures organizational data and operations but also protects modern society and the economy from the detrimental impact of cyber-attacks.

In Figure 6 six-stage maturity model shown above is the one that has been used in the quantification of cyber resilience processes in critical infrastructure settings. The posture of each level specifies another operating position, from Level 0 (Unfinished), where no security processes have been formalized, to Level 5 (Enhancement), where security processes are proactively reactive and continuously optimized. These stages have been titled Reportive, Reactive, Preventive, Detective, Responsive, and Adaptive, following the progression from basic reporting and reaction to active defense, detection, and long-term resilience. This maturity transition allows organizations to identify where they are today and balance high-priority development in particular areas throughout security functions as a basis for the scoring model described in the following section.

## 5. Discussion

### 5.1. Resilience Grading

Security strength assessments are a crucial step in measuring an organization’s readiness to face cyber threats [140,141]. This assessment process utilizes a standard grading scale to measure the extent to which the organization has achieved its security goals. The following is a deeper explanation of the grading scale and its implementation:**D (Did Not Meet)**: If a security element or achievement receives a “D” grade, this indicates that the element has not yet reached its goal. A “D” grade indicates that the achievement of this element is in the range of 0% to 20%. This is a concerning point, as it signifies significant weaknesses in that element. Organizations should promptly identify and address these weaknesses to achieve an adequate level of security.**A (Approaching)**: When an element or achievement is graded as “A”, this indicates that the element is approaching its goal but still falls within the range of over 20% to 50%. This shows progress, but there is still work required to reach the desired level of security strength. Organizations need to make further improvements to achieve an adequate level of security.**M (Moderately Met)**: The “M” grade indicates that an element or achievement has been moderately met, with achievement in the range of more than 50% to 75%. This is a positive sign that the organization has made significant progress in achieving better security. However, there is room for further improvement to reach the optimal level of security.**W (Well Achieved)**: The “W” grade signifies that an element or achievement has been well achieved, with achievement in the range of more than 75% to 90%. This is a commendable level of security strength but still allows for minor improvements. Organizations need to monitor this element to maintain a good level of security.**E (Exceeds Expectations)**: When an element or achievement receives an “E” grade, this means that the element has not only reached but exceeded expectations, achieving a very high level of security in the range of more than 90% to 100%. This is an outstanding achievement that demonstrates an organization’s ability to maintain security at the highest level. It is essential to continue monitoring and maintaining this very high level of security.

Additionally, resilience grading is represented in Table 4, providing a visual depiction of the resilience assessment and the extent to which an organization has attained its security objectives.

The importance of security strength assessments lies in the fact that it is a continuous process. Organizations must periodically reassess to monitor changes in the cybersecurity environment and ensure that all security elements remain adequate. In this process, there should be no significant weaknesses related to the assessed attributes. Consistency is crucial in determining the assigned grades, as described in Table 4 regarding assessments in terms of the percentages achieved. Assessors use this scale to determine the level of capabilities achieved. By consistently applying these criteria, each assessment can be based on a structured formality level. This not only allows for comparisons across an organization but also across different companies. Thus, this assessment process becomes a crucial tool to ensure the security and efficiency of organizations.

### 5.2. The Future of Cyber Resilience

The future of cybersecurity stands at a challenging crossroads, where technology advances rapidly amid increasingly sophisticated cyber threats. Several key aspects, supported by research [142,143], will shape this future, including the integration of artificial intelligence (AI) and machine learning (ML) technologies into cybersecurity defense. These advancements enable real-time data analysis, pattern recognition, and high-accuracy anomaly detection, allowing organizations to rapidly detect and efficiently respond to threats. Enhanced collaboration, as highlighted by sources [144,145], is imperative in the face of global cyber threats. International cooperation, information exchange, and partnerships between the public and private sectors will play a crucial role in combating the ever-growing complexity of cyber threats. As threats become more diverse, continuous education and training for cybersecurity professionals [146] will be a priority, ensuring their knowledge aligns with the latest trends and technologies. The expanding deployment of the Internet of Things (IoT), as noted by sources [147,148], poses a significant challenge for cybersecurity. Securing connected devices and the data generated will be essential for protecting critical infrastructure and sensitive information. The leadership within organizations, as emphasized in studies [149,150], will also impact security culture, with C-suite executives and boards of directors needing to prioritize cybersecurity and foster a security-conscious environment. Remaining vigilant against emerging threats, as stressed by sources [151], is essential, requiring organizations to continuously monitor and understand the tactics used by cyber-attackers. Stricter regulations [98], emphasizing data protection and privacy, will further shape the future cybersecurity landscape. Quantum computing [152], while it offers great potential, presents unique challenges, requiring ongoing preparation for the post-quantum cryptography era. The increasing number of internet-connected devices, as highlighted in studies [153,154], makes the IoT a vulnerability point for cyber threats. Efforts to secure the IoT ecosystem will be crucial to protect user data and privacy. Ransomware threats [155] continue to evolve and become more sophisticated, necessitating stronger defenses and effective recovery plans.

The integration of artificial intelligence into cyber-attacks, as discussed by researchers in [156], poses a future battleground between protective and attacking AI. Cyber-attacks by foreign states [157] are becoming more complex and coordinated, emphasizing the importance of national and international cybersecurity defense. Post-quantum security measures [69,158] are essential to keep data and communication secure in the era of quantum computing. The future will demand more skilled human resources in cybersecurity, with investments in training employees [159,160] in threat recognition, security analysis, and attack tactics. Increasingly strict cybersecurity regulations [98,161] will require organizations to adhere to guidelines in different jurisdictions. The research and innovation in cybersecurity [161,162] will continue to evolve, requiring organizations to invest in understanding new threats and creating innovative solutions. Public awareness of cybersecurity [163,164] will increase, placing pressure on organizations to maintain their reputation and customer privacy. Collaboration and information sharing between organizations [165] will become increasingly important for early warnings and rapid responses. Cloud security [16,166,167] will be a significant focus as the use of cloud services continues to grow. In conclusion, the future of cybersecurity, supported by various studies and sources, demands a deep understanding of these trends and strategic planning. Investments in security technology, human resource training, and collaboration between the public and private sectors will be crucial for organizations to maintain their resilience against evolving cyber threats [168,169,170].

Meanwhile, the model depicts current standards like ISO/IEC 15504, COBIT, and NIST CSF, and its unique structure and classification offer a novel approach to visualizing and measuring cyber resilience capability. This paper is especially helpful to energy, transport, healthcare, and other vital industry stakeholders who wish to benchmark and improve their security measures. Additional research could include empirical verification of the framework through simulation or case studies and its use in sector specifications. Finally, this research forms the basis of a stronger and adaptive approach to protecting critical systems in the era of the internet. Post-quantum cryptography (PQC) implementation is a new requirement in response to quantum computer threats to current cryptographic algorithms such as RSA and ECC. In the InfraGuard system, PQC is the next security control to be introduced under the security assurance component at the upper maturity levels (Level 4 and 5). Quantum-resistant primitives approved by NIST, such as CRYSTALS-Kyber key encapsulation and Dilithium digital signatures, can be applied in secure communication protocols and data protection systems. Quantum-resistant systems also tend to require more processing power and memory, though, which will affect the system’s latency and computational overhead, especially in legacy or embedded systems. Therefore, PQC implementations must be piloted phase-wise under test conditions in tandem with the infrastructure capacity to ensure both the scope for recovery and operational capacity.

### 5.3. Exploratory Scenarios for the Framework’s Application

To illustrate the use of the InfraGuard Cybersecurity Framework in real practice, three exploratory case scenarios are presented, one for each discrete critical infrastructure sector. The fictional scenarios simulate high-impact cyber-attacks and show the use of the framework in assessing organizational resilience. While not realistic, the scenarios are worded based on extensively publicized attack methods and operational vulnerabilities seen in real-world infrastructure environments. All accounts emphasize the incident attributes, technical vulnerabilities, relevant dimensions of resilience, and normal levels of maturity. And in Table 5 summary of all scenarios.

**Scenario 1: National Electrical Grid Disruption**—A cyber-attack on an electric grid’s state-owned SCADA systems initiates widespread regional blackouts. Modbus TCP/IP-based SCADA systems have no encryption and authentication activity and thus are susceptible to command injections and session hijacking. Poor network segmentation design facilitates easy lateral movement between operating zones. There are no inventories of assets or Security Information and Event Management (SIEM) solutions, and the recovery is manual within 24 h. This falls within the terms of the technical vulnerabilities exploited in previous attacks such as the 2015 Ukraine grid attack.

**Key Impacted Components**: Situational awareness, risk management, active defense;**Indicative Resilience Level**: Very low (Level 1).

**Scenario 2: Ransomware in a Smart Hospital System**—Infection of a metropolitan hospital network with ransomware encrypts electronic health records and cripples IoT-enabled medical equipment. The segmentation at the hospital is minimal, with shared access between administrative workstations and clinical systems. There is no active and functional incident response mechanism where endpoint protection is invoked. The 12 h recovery causes temporary disruption to critical care unit processes. This is the type of exposure that has been exploited in real attacks such as the WannaCry attacks against healthcare networks.

**Key Impacted Components**: Preparedness, infrastructure resilience, incident recovery;**Indicative Resilience Level**: Developing (Level 2–3).

**Scenario 3: An Airport Cyber Sabotage Incident**—There is a cyber-attack on the flight coordination and baggage handling processes at an international airport. The ISO/IEC 27001 certified airport is being centrally monitored by a SOC (Security Operations Center) without live cyber exercises or red team exercises between departments. The baggage system operates legacy PLCs with proprietary, unpatched firmware and is under supply chain compromise or insider exploitation. It can be recovered within 5 h, but the after-incident analysis determines that there is no consolidation of the protocols between the IT and OT teams.

**Key Impacted Components**: Infrastructure preparedness, active defense, response integration;**Indicative Resilience Level**: Strong (Level 3–4).

While conceptual in nature, these vignettes are representative of potential real-world situations and illustrate key technical deficiencies generic to critical infrastructure. Empirical confirmation through forensic analyses, red teaming, and formal interviewing with subject matter experts must be incorporated into follow-up research to verify the robustness and effectiveness of the framework in use.

### 5.4. Technological Integration and Practical Relevance

The proposed framework is centered on the strategic use of new technologies such as artificial intelligence (AI), machine learning (ML), and automated threat detection tools. These are the primary technologies that facilitate predictive monitoring, anomaly detection, and real-time incident response. For example, AI-driven analytics can be employed to analyze log data and identify potential threats based on behavioral patterns. The scoring framework of this model allows this through prioritizing above-average rankings to organizations that employ real-time monitoring and automated mitigation paradigms, thus allowing for flexibility towards dynamic threats. The employment of technologies such as AI-CRM and system-theoretic techniques such as STPA-Sec is natively built into the framework’s design, which aligns with models that possess proactive threat modeling and adaptive learning abilities.

Aside from technical value, the InfraGuard framework is of pragmatic and strategic use to critical infrastructure leaders in the public and private sectors. Through its breakdown of resilience into quantifiable elements and mapping against specific maturity levels, leaders can identify gaps, spend appropriately, and benchmark the performance against industry standards such as NIST CSF, ISO/IEC 27001, and COBIT. This structural approach offers channels for technical departments and management to communicate, through which cybersecurity policies are aligned with organizational objectives, as well as conformity needs. In this, the model is not just an assessment tool but also an indicator for policy development, budgeting, and cyber resilience capacity building. For the real-world application of threat detection within the InfraGuard system, different AI/ML models may be applied depending on the infrastructure situation. Anomaly detection from logs, for example, may employ Isolation Forests or Autoencoders, while time-series traffic forecasting in network analyses can operate with LSTM-based recurrent neural networks. The training data may include system event logs, IDS/IPS alert logs, and network packet captures which are normalized and cleansed of noise through preprocessing. Integration with existing infrastructure may be achieved via modular detection engines at the network edge or within SIEM platforms. Benchmark performance metrics such as the precision, recall, and false positive rate are critical to ensuring operational effectiveness, and countermeasures such as feedback loops, adaptive thresholding, and ensemble decision logic can be utilized to prevent spurious alarms and computational overload.

## 6. Conclusions

This study proposed a systematic approach to assessing the cyber resilience of critical infrastructure through the integration of capability maturity levels and domain-oriented security factors. In developing the InfraGuard Cybersecurity Framework, we offer a realistic model not only outlining the growth of security process maturity but also segmenting the measures of resilience into three strategic areas: Shield, Space, and Sword. These dimensions consist of levels that accommodate cybersecurity activities across a spectrum, ranging from monitoring and prevention through to readiness and recovery. The model offers conceptual definitions, as well as operational guidance, for organizations looking to enhance their resilience position in more complex threat environments.

By integrating multi-level security process maturity and functional areas reflective of different levels of cyber defense, a positive resilience model can be developed. The InfraGuard Cybersecurity Framework developed here enables organizations to score at six levels of maturity in terms of their resilience capability, from incomplete to highly sophisticated processes, considering different functions such as situational awareness, active defense, infrastructure readiness, and incident recovery. Ordering these parameters into a firm matrix, the model supports the proactive detection of threats, instantaneous responses, and a focused method to strengthen cyber resilience in an orderly fashion.

Predictions and decisions regarding preventive measures involve an analysis of historical threat trends, present process maturity, and social–technical preparedness, all of which are incorporated into the multilayered design of the framework. The inclusion of topics like “Cyber as a Shield” and “Cyber as a Space” lays particular focus on initial discovery, risk management, and infrastructure awareness, which are drivers of potential vulnerabilities. These can be used by organizations to give high priority to modernization, develop scenario-dependent backup plans, and implement technology like AI-based monitoring in an effort to avoid failures before a significant disruption occurs.

This study implies levels of resilience grading and performance goals drawn from an official evaluation of key indicators by geography. Each process is segmented into levels (0 to 5) and correlated with tangible, evident signs: the recovery period, technology take-up, and operational uniformity. Companies can measure themselves in their present status against leading-class examples and best practices like NIST CSF or AI-CRM. Even without much data, simulated conditions and qualitative analyses can be used within the model to enable comparative assessments that identify capability gaps and can be fed back to improve the desired domains.

This study provides a graphical and structured method that helps decision-makers attain organizational cyber resilience in both strategy and operations. By synchronizing the security capability across maturity levels and domains, managers can more effectively utilize resources, make policies, and plan for employee training. In addition, the framework’s capability to map across different slices of infrastructure and adherence to global standards ensures that it is an effective guide to global cooperation, encouraging a collective effort towards improving cyber defense and minimizing the broader societal and economic impacts of cyber-attacks.

Despite the formal process outlined in this study, there remain some limitations. Empirical testing of the model in various categories of critical infrastructure, i.e., energy, transportation, or health systems, has not been performed. Consequently, its transferability across domain-specific operating environments and applicability need to be explored further through cross-sector case studies or field tests. In addition, the model fails to adequately address the issues of deployment in real heterogenous systems, like technical interoperability, performance penalties, and resource needs—issues that would affect its deployment, particularly in legacy or hybrid environments. It is anticipated that follow-up studies would broaden the scope of this framework by experimentally validating its applicability in real environments and by incorporating deployment models that are system-compatibility- and operational-feasibility-aware. Another limitation lies in the practical implementation of this framework across heterogeneous environments. Legacy infrastructure, performance constraints, and complex integration can hinder real-time monitoring and automation. Future research should explore scalable solutions that balance security visibility with operational efficiency, particularly in latency-sensitive systems. This study sets the groundwork for a comprehensive validation process, which will involve empirical testing through domain-specific case studies and expert evaluations to confirm the framework’s practical applicability and technical soundness.

## Figures and Tables

**Figure 1 sensors-25-04545-f001:**
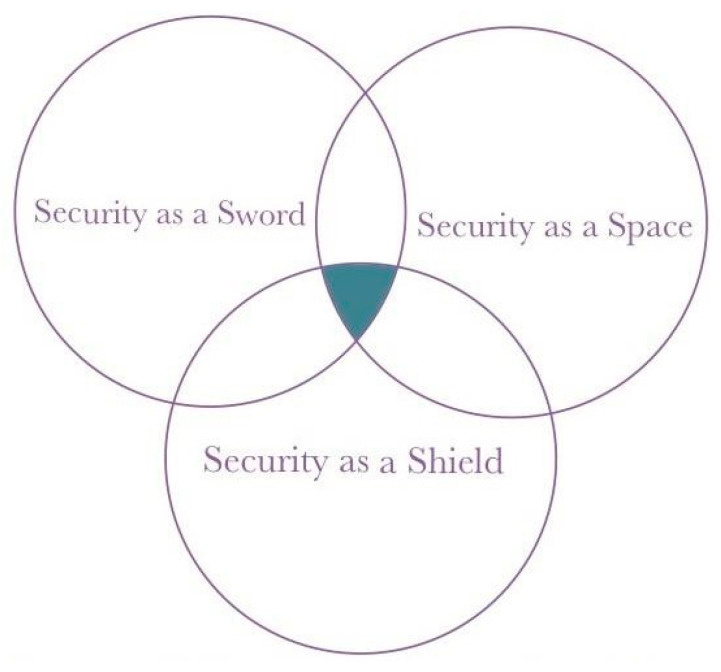
Classification of cybersecurity approaches [116].

**Figure 2 sensors-25-04545-f002:**
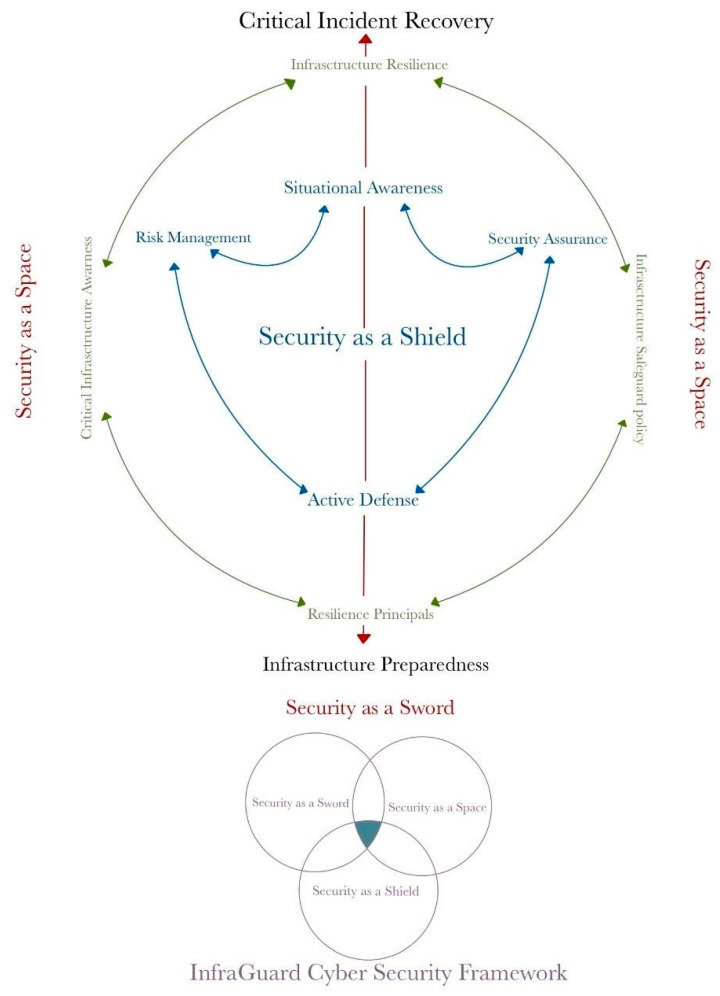
InfraGuard Cybersecurity Framework [116].

**Figure 3 sensors-25-04545-f003:**
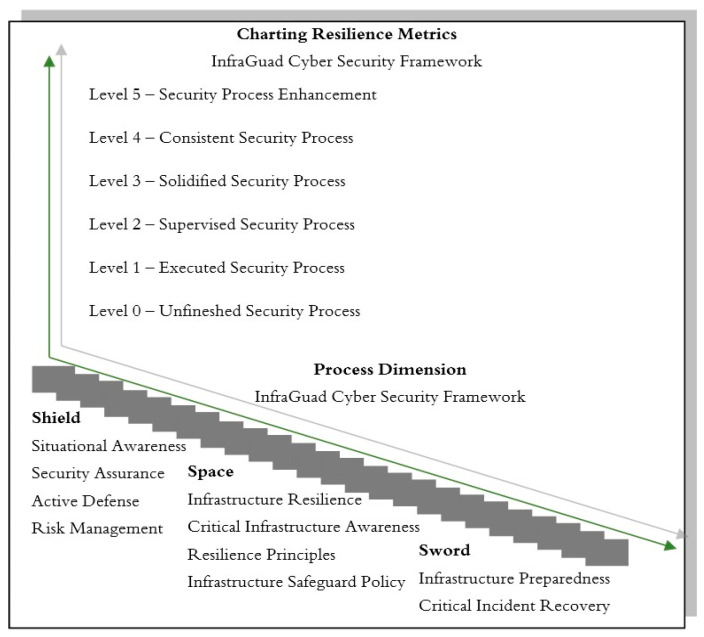
Charting resilience metrics.

**Figure 4 sensors-25-04545-f004:**
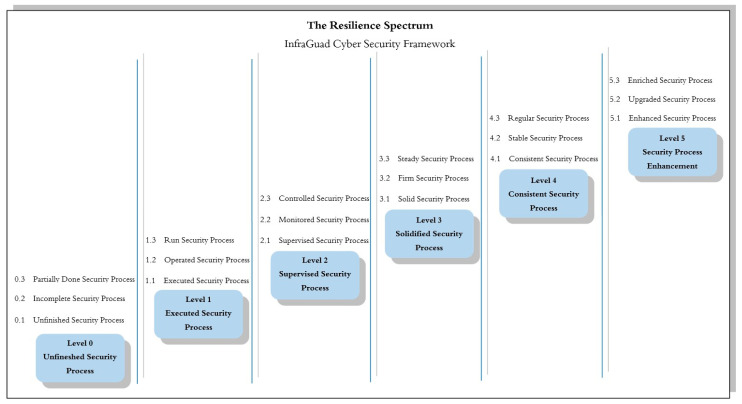
The resilience spectrum.

**Figure 5 sensors-25-04545-f005:**
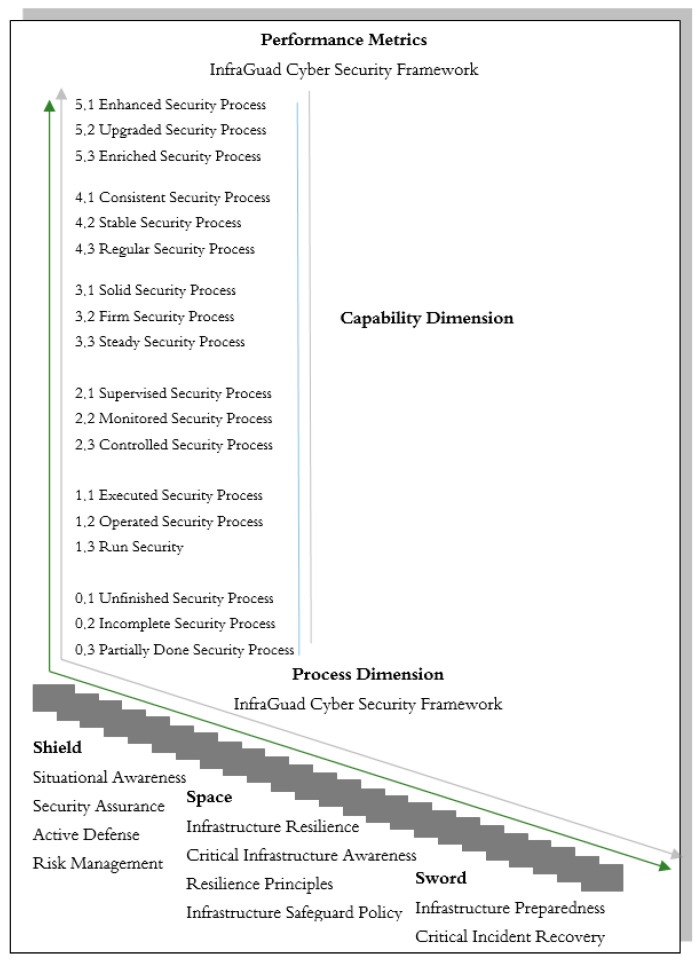
Performance metrics.

**Figure 6 sensors-25-04545-f006:**
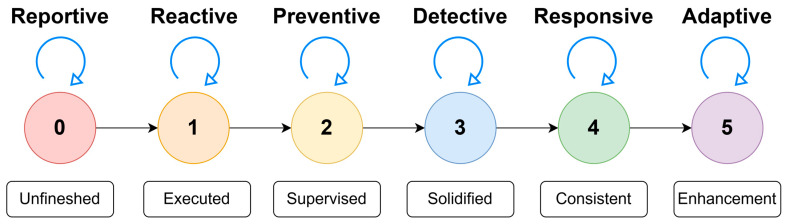
Maturity levels in cyber resilience process development.

**Table 1 sensors-25-04545-t001:** Domains and components in the cybersecurity framework for critical infrastructure.

Domain	Component	Indicator	Description	Reference
Cyber as a Shield	Situational Awareness	Threat detection and monitoring capabilities	The organization can proactively observe operational changes and identify potential cyber threats.	[30,95,96,116,117,118]
	Security Assurance	Risk assessments and security controls	Includes routine evaluations and the enforcement of strict standards to ensure system protection.	
	Active Defense	Rapid response to threats	Involves using tools and strategies to detect and prevent attacks before system damage occurs.	
	Risk Management	Risk identification and mitigation	A systematic process to assess threats and prioritize mitigation actions.	
Cyber as a Space	Infrastructure Resilience	System robustness and recovery capabilities	Infrastructure can sustain operation and recover during or after cyber incidents.	[30,89,90,91,92,93,94,116]
	Critical Infrastructure Awareness	Organizational awareness of vital systems	A deep understanding of the infrastructure’s national significance and associated risks.	
	Resilience Principles	Resilient design and operational philosophy	Foundational principles for building systems that can withstand disruptions.	
	Infrastructure Safeguard Policy	Protective policies and procedures	Formal documents and procedures to secure physical and digital infrastructure from threats.	
Cyber as a Sword	Infrastructure Preparedness	Pre-emptive readiness and training	The presence of incident response plans, personnel training, and scenario simulations.	[30,111,112,113,114,115,116]
	Critical Incident Recovery	Recovery speed and continuity measures	The ability to restore system functions quickly and efficiently after disruptions.	

**Table 2 sensors-25-04545-t002:** Quantitative indicators and scoring for each framework component.

Component	Indicator	Measurement Criteria	Description
Situational Awareness	- % of systems with real-time monitoring - Mean detection time (MTTD)	Based on coverage of monitoring and average time to detect anomalies	0: No monitoring system 1: Manual observation only 2: Partial system monitoring 3: Full system monitored periodically 4: Real-time monitoring 5: Real-time + automated anomaly detection with alerting
Security Assurance	- Number of security controls implemented - Certification status	Refer to implemented frameworks (e.g., ISO 27001) and documented controls	0: No controls or certifications 1: Basic firewall/AV only 2: Partial controls implemented 3: Formal internal policy with controls 4: Certification in progress 5: Full certification (e.g., ISO 27001) and up-to-date controls
Active Defense	- Mean Time to Detect/Respond (MTTD/MTTR) - Number of false positives	Based on system responsiveness and detection accuracy	0: No response capability 1: A delayed manual response (>72 h) 2: Manual monitoring, reactive response 3: Semi-automated alerts and mitigation 4: Full incident response plan with automation 5: Automated detection and active defense with <2% false positives
Risk Management	- Frequency of risk assessments - % of mitigated high-risk items	Based on risk governance process and follow-up	0: No risk assessment 1: Ad hoc assessments only 2: Annual risk assessments 3: Quarterly assessments 4: Documented mitigation tracking 5: Continuous risk analysis with >90% risk mitigation execution
Infrastructure Resilience	- System uptime (% availability) - Maximum downtime per year	Based on service continuity and fault tolerance	0: Unstable system, frequent failures 1: Downtime of >48 h/year 2: Downtime of 24–48 h/year 3: Downtime of 8–24 h/year 4: Downtime of <8 h/year 5: High-availability setup with <1 h/year downtime
Critical Infrastructure Awareness	- % of critical assets identified and classified - Availability of critical asset inventory	Based on documentation and prioritization	0: No asset classification 1: Initial asset list only 2: Incomplete inventory 3: Full classification but outdated 4: An up-to-date list of critical systems 5: Inventory integrated with risk and threat modeling tools
Resilience Principles	- Implementation of redundancy, backup, and failover systems	Based on architectural design and redundancy coverage	0: No resilience mechanisms 1: Manual backups only 2: Periodic backups and isolated recovery plans 3: Redundant systems in core infrastructure 4: Partial failover capability 5: Full redundancy and automated failover across systems
Infrastructure Safeguard Policy	- Number of security-related policies - Frequency of policy updates	Based on comprehensiveness and relevance of official policy documents	0: No formal policies 1: Single general policy 2: Multiple but outdated policies 3: Up-to-date, role-specific policies 4: Policies reviewed annually 5: Integrated, reviewed biannually and aligned with national/international standards
Infrastructure Preparedness	- Frequency of cyber drills - % of trained personnel	Based on preparedness programs and regular testing	0: No training or drills 1: Basic training for some staff 2: Annual training for IT team 3: Annual drills across departments 4: Semi-annual simulations 5: Full organization involved in quarterly simulations with >90% personnel trained
Critical Incident Recovery	- Mean Time to Recovery (MTTR) - % of services restored within SLA	Based on recovery performance and SLA compliance	0: Recovery not defined 1: MTTR of >72 h 2: MTTR of 48–72 h 3: MTTR of 24–48 h 4: MTTR of 4–24 h 5: MTTR of <4 h, 100% SLA compliance

**Table 3 sensors-25-04545-t003:** Total resilience score.

Score	Interpretation
0–20	Low Resilience
21–35	Developing Resilience
36–45	Strong Resilience
46–50	Optimized and Adaptive Resilience

**Table 4 sensors-25-04545-t004:** Resilience grading.

Resilience Grading		
Abbreviation	Description	% Achieved
D	Did Not Meet	0–20% achievement
A	Approaching	>20–50% achievement
M	Moderately Met	>50–75% achievement
W	Well Achieved	>75–90% achievement
E	Exceeds Expectations	>90–100% achievement

**Table 5 sensors-25-04545-t005:** Summary of scenario-based framework application.

Scenario	Sector	Main Incident	Technical Notes	Key Components	Resilience Level
Electrical Grid Disruption	Energy (Power Grid)	SCADA-targeted cyber-attack	Modbus TCP/IP, no encryption, flat network, manual recovery	Situational Awareness, Risk Management, Active Defense	Very Low (Level 1)
Smart Hospital Ransomware	Healthcare	Ransomware and medical IoT disruption	Weak segmentation, no IR coordination, outdated backups	Preparedness, Resilience, Incident Recovery	Developing (Level 2–3)
Airport System Sabotage	Transportation	System outage via OT compromise	Legacy PLCs, SOC present, no unified IT-OT drills	Preparedness, Defense, Response Coordination	Strong (Level 3–4)

## Data Availability

The data can be made available on request.

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
