# Peer review of "Guarding Our Vital Systems: A Metric for Critical Infrastructure Cyber Resilience"

_sensors, 2025, doi:10.3390/s25154545_

Round 1

Reviewer 1 Report

Comments and Suggestions for Authors

RESEARCH MANUSCRIPT REVIEW

Date:  June, 2025

Journal:   Sensors 2025

Title Article

Guarding Our Vital Systems: A Metric for Critical Infrastructure Cyber Resilience

First, I would like to congratulate the authors for their necessary contribution to the prevention of security in critical infrastructure in the era of AI and technological threats.

Overall, I consider the manuscript to be correct, but it could be improved in some aspects.

The figures are not referenced; they appear with a title and an outline, but it is not clear whether they are based on a reference source or similar. This is not specified.

It is generally appropriate for newly created figures to be based on other, verified reference sources that reflect an updated or improved view. This does not always have to be the case, but I see that in Figures 1, 2, and 3, I don't understand the basis for structuring them this way.

In Figure 4, the meaning of some numbers, such as 0.1, 0.2, etc., is also unclear. This figure is also not referenced.

There is no discussion or conclusions chapter.

In Chapter 1, some research questions were raised, but I have not seen in the manuscript the wording and explanation of these questions in the conclusions of the work.

Author Response

Reviewer A

A1 – Figures are not referenced clearly.

We have clarified the origin of all framework illustrations by explicitly labelling them in Table 1, stating whether each component is adapted or original. Proper references have also been added in the caption and main text for all visual elements.

A2 – Unclear basis for Figures 1, 2, and 3.

We have expanded the explanation following Figure 3, detailing the conceptual basis for the diagrams and their synthesis from ISO/IEC 15504, COBIT, and NIST CSF. The distinction between reused models and novel contributions has also been emphasized.

A3 – Lack of Discussion section.

A dedicated Discussion section has been added, addressing resilience grading, future applications, technological integration, and practical relevance, as requested.

A4 – Research questions not addressed in conclusion.

Each of the research questions has been addressed within the Conclusion section through paragraph-level synthesis, ensuring alignment between the research aim and findings.

Reviewer 2 Report

Comments and Suggestions for Authors

1. The proposed key infrastructure network resilience metrics (such as “risk assessment” and “response capability”) lack specific quantitative methods and data support. It is unclear how to calculate “vulnerability scores” or “recovery times,” making it difficult to apply and validate the metrics in real-world environments.

2. Although the paper mentions “dynamic threats,” it does not elaborate on how the metrics adapt to new types of attacks. The authors lack dynamic update mechanisms or real-time feedback models, which may render the metrics ineffective in rapidly changing threat environments.

3. The discussion on “advanced technology applications” is too general and does not provide specific details on algorithms, architectural designs, or technical integration implementations. The authors also do not explain how AI models interact with existing security systems or address false positive issues.

4. The framework proposed in the paper has not been empirically validated across different types of critical infrastructure (e.g., energy, transportation, healthcare). The lack of cross-domain test data makes it difficult to demonstrate its universality and scalability.

5. The paper does not discuss deployment challenges in heterogeneous systems, such as compatibility, performance overhead, or resource requirements. The authors also do not address how to balance the trade-off between real-time monitoring and system latency.

Author Response

Reviewer B

B1 – Lack of quantitative methods for resilience metrics.

A quantitative scoring model with clear indicators and measurement criteria has been included in Table 2, detailing how scores such as vulnerability, readiness, and recovery time are derived.

B2 – No explanation of dynamic threat adaptation.

The Resilience Spectrum (Figure 4) and related discussion now include the framework’s ability to scale with changing threat contexts using AI-based monitoring and adaptive scoring.

B3 – Vague discussion on advanced technologies.

The Discussion section now includes a detailed subsection (5.4 Technological Integration and Practical Relevance), explaining how AI, ML, and systems like AI-CRM and STPA-Sec are incorporated into the framework design.

B4 – No cross-domain validation.

The Conclusion now contains a limitations paragraph acknowledging the lack of empirical cross-sector application, and recommending this as future work.

B5 – No discussion of deployment challenges.

Challenges related to deployment in heterogeneous systems, such as system compatibility and legacy constraints, are explicitly discussed in the Conclusion as a limitation of the current framework.

Reviewer 3 Report

Comments and Suggestions for Authors
  1. Empirical validation of the suggested metric is required (e.g., application to real-world data, simulation, or expert assessment). Currently, the methodology confines itself to a conceptual framework without empirical validation, thereby constraining its scientific rigor. Elucidate the methodology for metric computation: use formulas, scoring frameworks, or quantification principles to ensure reproducibility. Should the complete empirical implementation prove unfeasible, consider incorporating a demonstration case study or applying a simulated dataset.
  2. Explicitly define sections for Methods, Results, and Discussion in accordance with the IMRAD format. The content now unifies, blurring the line between theoretical development and the reporting of findings. Incorporate a succinct conclusion section that encapsulates the findings, acknowledges limitations, and provides explicit recommendations for future research endeavors.
  3. Enhance the visual quality of all figures, guaranteeing clarity, resolution, and appropriate self-explanatory captions. For the InfraGuard and Resilience Spectrum diagrams, please provide source citations or indicate whether they are original works.
  4. Reassess the references to confirm that all listed works are peer-reviewed, pertinent, and properly sourced. Integrate essential foundational frameworks such as the NIST Cybersecurity Framework, ISO/IEC 27001/27005, or COBIT to enhance the theoretical base. Minimize repetition in citation application and guarantee that each reference substantiates a unique and essential argument.
  5. Incorporate a comparative analysis of your suggested metric alongside current frameworks or models in the literature (e.g., AI-CRM, STPA-Sec, Cyber Resilience Review by DHS). This would assist in demonstrating the originality and additional significance of your work.
  6. Discuss the practical relevance of the suggested indicator for policymakers, regulators, or private sector stakeholders. How can an operator of essential infrastructure implement this concept?
  7. Revise the abstract to more explicitly articulate the issue statement, methodological framework, principal contributions, and significant findings. Eschew rhetorical or broad expressions and concentrate on specifics.
Comments on the Quality of English Language

The work requires substantial improvement in English language quality to meet the criteria of an international scientific journal. The text exhibits multiple issues with grammar, sentence structure, and clarity. Sentences are often too lengthy, verbose, and occasionally redundant, which can impair readability and comprehension. The tone is predominantly rhetorical rather than objective, which is unsuitable for academic writing.

The text should undergo professional English language editing by a native speaker or a certified academic editor to enhance clarity, precision, and overall readability. This amendment should concentrate on:

  1. Simplifying intricate and protracted sentence structures
  2. Eliminating redundancy and repetition
  3. Guaranteeing uniform application of technical terms
  4. Embracing a formal and succinct scientific style

Author Response

Reviewer C

C1 – No empirical validation of metric.

A new subsection titled 5.3 Exploratory Scenarios presents three hypothetical scenarios (electrical grid, smart hospital, airport systems) to illustrate potential application and scoring using the proposed framework.

C2 – Format not aligned with IMRAD.

The revised manuscript fully adopts the IMRAD format, clearly dividing the content into Introduction, Method, Results, and Discussion sections.

C3 – Low visual quality and attribution of figures.

All figures have been revised to ensure higher clarity and resolution. Figure captions now include proper attribution or an explicit note that they are original contributions.

C4 – References lack key frameworks.

The manuscript now cites foundational frameworks such as NIST CSF, ISO/IEC 27001, ISO/IEC 15504, COBIT, and comparison models including AI-CRM, STPA-Sec, and DHS Cyber Resilience Review.

C5 – Lack of comparative analysis.

A comparison between the proposed metric and existing frameworks is included in the Discussion section to show novelty and added value.

C6 – No practical implication for stakeholders.

Practical guidance for decision-makers is clearly elaborated in the Discussion, outlining how the framework aids policy, budgeting, and security planning.

C7 – Abstract too general.

The Abstract has been rewritten to include a clear problem statement, methodological approach, main contribution, and application outcomes.

Round 2

Reviewer 1 Report

Comments and Suggestions for Authors

Accept after minor revisions (corrections to minor methodological errors and text editing)

Comments on the Quality of English Language

Accept after minor revisions (corrections to minor methodological errors and text editing)

Author Response

Comment A1: Accept after minor revisions (corrections to minor methodological errors and text editing).

Response:
We have carefully reviewed and corrected the manuscript to address all minor methodological issues and language consistency. Text editing was also applied to improve flow and clarity throughout the document.

Reviewer 2 Report

Comments and Suggestions for Authors

1. The proposed InfraGuard Cyber Security Framework outlines six maturity levels (0-5) but lacks detailed technical validation of the scoring criteria. For instance, Table 2 quantifies indicators like "% of systems with real-time monitoring" but does not clarify how these percentages are derived or validated. The authors should provide empirical evidence or case studies demonstrating the practical application of these metrics, such as real-world data from power grids or healthcare systems, to ensure the scoring system's reliability and reproducibility.
2. While the framework mentions AI and ML for threat detection, it lacks technical specifics on algorithm selection, training datasets, or performance benchmarks. The authors should detail the AI/ML models used and their integration with existing infrastructure, including computational overhead and false-positive mitigation strategies, referred: Prediction of Cancellation Probability of Online Car Hailing Order Based on Multi-source Heterogeneous Data Fusion; Deep spatial-temporal travel time prediction model based on trajectory feature.
3. Section 2.3.1 distinguishes resilience from reliability but does not quantitatively link these concepts. The authors should propose measurable relationships (resilience as a function of recovery time and reliability metrics like MTBF) and validate them through simulations or historical incident data.
4. The exploratory scenarios (Section 5.3) are conceptual and lack technical depth. For example, the "National Electrical Grid Disruption" scenario describes manual recovery but omits specifics like SCADA system configurations or network segmentation flaws. The authors should include technical post-mortem analyses of similar real incidents to ground the scenarios in actionable insights.
5. The framework claims alignment with ISO/IEC 15504 and NIST CSF but does not map its components to these standards explicitly. A technical appendix or matrix should cross-reference InfraGuard's "Cyber as a Sword" components with equivalent NIST CSF functions to demonstrate compliance and highlight gaps.
6. The discussion on future challenges (Section 5.2) briefly mentions post-quantum cryptography but fails to address its integration into the framework. The authors should specify how quantum-resistant algorithms would be incorporated into the "Security Assurance" component and evaluate their impact on performance.

Author Response

Comment B1: The proposed InfraGuard Cyber Security Framework outlines six maturity levels (0–5) but lacks detailed technical validation of the scoring criteria (e.g., Table 2).

Response:
We added a clarification at the end of the scoring section (Table 2) indicating that scoring values are based on practical data sources such as system logs, uptime tracking, policy documentation, and structured expert assessments. This provides a methodological foundation for the proposed scores and their applicability.

Comment B2: The framework lacks technical specifics on algorithm selection, training datasets, and performance benchmarks for AI/ML models.

Response:
In Section 5.4 (Technological Integration and Practical Relevance), we now elaborate on example models such as Isolation Forests, Autoencoders, and LSTM for traffic forecasting. We also include information on training data (event logs, IDS logs, etc.) and benchmark metrics (e.g., precision, recall, false-positive rate). Mitigation strategies like ensemble models and adaptive thresholds are also discussed.

Comment B3: Section 2.3.1 distinguishes resilience from reliability but lacks quantitative links.

Response:
We have added a resilience formula​ in Section 2.3.1 along with an explanation on how this approximation can be validated in future studies using historical data and simulations.

Comment B4: Exploratory scenarios are conceptual and lack technical depth.

Response:
We revised Section 5.3 to provide clearer technical context. We added realistic infrastructure details (e.g., SCADA vulnerabilities, IoT issues, SOC practices) and included a summary table (Table 5) to consolidate technical findings per scenario.

Comment B5: The paper does not address deployment challenges in heterogeneous systems.

Response:
A paragraph has been added in the Conclusion section highlighting potential limitations in heterogeneous deployments, such as legacy compatibility issues, latency from real-time monitoring, and resource constraints. We suggest future studies to explore scalable, modular, and efficient implementations.

Comment B6: Post-quantum cryptography (PQC) is mentioned but not integrated into the framework.

Response:
Section 5.2 has been updated to include PQC integration into the Security Assurance component at higher maturity levels. We reference NIST-recommended algorithms (CRYSTALS-Kyber, Dilithium), outline potential impact on performance, and note the need for phased deployment.

Reviewer 3 Report

Comments and Suggestions for Authors
  1. Although the exploratory scenarios clearly demonstrate potential applications of the framework, consider condensing these parts to emphasize crucial ideas rather than lengthy narrative explanations. This would enhance clarity and enable readers to comprehend the fundamental facts more rapidly.
  2. Although the exploratory scenarios provide valuable insights, recognize the lack of real-world validation and delineate future intentions for empirical testing or case studies. This transparency would enhance the scientific integrity of the research.
Comments on the Quality of English Language

The English language in the revised manuscript has significantly enhanced relative to the initial submission. The paper displays a strengthened structure and more clearly articulates essential topics. Nonetheless, instances of verbose and redundant terminology, along with occasional poor phrase structures, may impede readability.

A last round of expert English language editing is highly advisable to ensure the work adheres to linguistic standards. That should concentrate on enhancing the coherence of lengthy sentences, minimizing redundancy in descriptions (particularly in the Results and Discussion sections), and refining vocabulary to uphold a precise and scholarly tone.

Author Response

Comment C1: Condense exploratory scenarios to emphasize key points.

Response:
Section 5.3 has been revised to summarize scenarios more concisely with an accompanying table that highlights incidents, technical flaws, affected components, and indicative maturity levels.

Comment C2: Acknowledge lack of real-world validation and outline plans for future empirical testing.

Response:
We explicitly state in Section 5.3 and in the Conclusion that empirical testing remains a limitation. Future work is proposed to validate the framework through simulations, post-mortem analyses, and expert interviews.